# Efficacy and Safety of Two-Drug Regimens with Dolutegravir plus Rilpivirine or Lamivudine in HIV-1 Virologically Suppressed People Living with HIV

**DOI:** 10.3390/v15040936

**Published:** 2023-04-10

**Authors:** Carlos Dueñas-Gutiérrez, Luis Buzón, Roberto Pedrero-Tomé, José A. Iribarren, Ignacio De los Santos, Sara De la Fuente, Guillermo Pousada, Miguel Angel Moran, Estela Moreno, Eva Ferreira, Julia Gómez, Jesús Troya

**Affiliations:** 1Infectious Diseases Division, Hospital Universitario Clínico de Valladolid, 47003 Valladolid, Spain; 2Infectious Diseases Division, Hospital Universitario de Burgos, 09006 Burgos, Spain; luisbuzonmartin78@gmail.com; 3Infanta Leonor University Hospital Research and Innovation Foundation, 28031 Madrid, Spain; robertopedrerotome@gmail.com; 4Infectious Diseases Department, Hospital Universitario de Donostia, 20014 San Sebastián, Spain; joseantonio.iribarrenloyarte@osakidetza.eus; 5Infectious Diseases Division, Hospital Universitario de La Princesa, 28006 Madrid, Spain; isantosg@hotmail.com; 6Infectious Diseases Division, Hospital Universitario Puerta de Hierro, 28222 Madrid, Spain; 7Infectious Diseases Division, Hospital Universitario de Txagorritxu, 01009 Vitoria, Spain; guille.pousada@gmail.com; 8Infectious Diseases Division, HospitalÁlvaro Cunqueiro, 36312 Vigo, Spain; miguelangelmoranrodriguez@gmail.com; 9Infectious Diseases Division, Complejo Hospitalario de Navarra, 31008 Pamplona, Spain; estela.moreno.garcia@navarra.es; 10Infectious Diseases Division, Hospital de Segovia, 47002 Segovia, Spain; eferreira@saludcastillayleon.es; 11Infectious Diseases Division, Hospital Universitario de Salamanca, 37007 Salamanca, Spain; jgomezb@saludcastillayleon.es; 12Internal Medicine Department, Hospital Universitario Infanta Leonor, 28031 Madrid, Spain; jestrogar@hotmail.com

**Keywords:** HIV/AIDS, 2DR, virologically suppressed patients

## Abstract

Background: The high effectiveness and safety of the two-drug (2DRs) strategy using dolutegravir (DTG) plus lamivudine (3TC) have led to international guidelines recommending their use for treatment-naive HIV patients. In virologically suppressed patients, de-escalating from 3DRs to DTG plus either rilpivirine (RPV) or 3TC has shown high rates of virological suppression. Objectives: This study aimed to compare the real-life data of two multicenter Spanish cohorts of PLWHIV treated with DTG plus 3TC (SPADE-3) or RPV (DORIPEX) as a switch strategy, not only in terms of virological suppression, safety, and durability but also in terms of immune restoration. The primary endpoint was the percentage of patients with virological suppression on DTG plus 3TC and DTG plus RPV at weeks 24 and 48. The secondary outcomes included the proportion of patients who experienced the protocol-defined loss of virological control by week 48; changes in immune status in terms of CD4+ and CD8+ T lymphocyte counts and the CD4+/CD8+ ratio; the rate, incidence, and reasons for discontinuation of treatment over the 48-week study period; and safety profiles at weeks 24 and 48. Methods: We conducted a retrospective, observational, multicenter study of 638 and 943 virologically suppressed HIV-1-infected patients in two cohorts who switched to 2DRs with DTG plus RPV or DTG plus 3TC. Results: The most frequent reasons for starting DTG-based 2DRs were treatment simplification/pill burden or drug decrease. The virological suppression rates were 96.9%, 97.4%, and 99.1% at weeks 24, 48, and 96, respectively. The proportion of patients with virological failure over the 48-week study period was 0.01%. Adverse drug reactions were uncommon. Patients treated with DTG+3TC increased CD4, CD8, and CD4/CD8 parameters at 24 and 48 weeks. Conclusions: We conclude that DTG-based 2DRs (combined with 3TC or RPV) in clinical practice were effective and safe as a switching strategy, with a low VF and high viral suppression rates. Both regimens were well tolerated, and ADR rates were low, including neurotoxicity and induced treatment discontinuations.

## 1. Introduction

International guidelines recommend the use of a three-drug (3DRs) combined antiretroviral therapy (cART) regimen as the standard of care for the treatment of most people living with HIV-1 (PLWHIV) [1,2,3,4]. This strategy has enabled the control of HIV-1 infection with efficacy rates above 90% [5], progressive immune system restoration, and significantly reduced acquired immunodeficiency syndrome (AIDS) events and other complications associated with HIV-1 infection. The excellent efficacy and safety of two- drug (2DRs) strategies demonstrated in clinical trials have led to international guidelines to change their recommendations to include the use of dolutegravir (DTG)-based 2DRs plus lamivudine [3TC] for treatment-naive HIV patients [1,2,3,6,7,8]. No emergent resistant virus to dolutegravir has ever been reported in clinical trials of patients for whom dolutegravir was prescribed in the context of such two-drug regimens [9,10]. In virologically suppressed patients, de-escalating from 3DRs to DTG plus either rilpivirine (RPV) or 3TC has shown high rates of virological suppression and safety [11,12,13,14]. A recent meta-analysis showed that DTG-based 2DR successfully kept virological control at 48 weeks, as only 0.7% of patients experienced viral failure, and there were no cases of emerging DTG resistance. In addition, only one patient had a primary RPV resistance mutation [15].

Low CD4/CD8 ratios have been associated with T-cell activation, immune senescence, and higher morbidity and mortality, mainly related to the more frequent occurrence of non-AIDS events [16,17]. However, data regarding the impact of 2DRs on immune activation and inflammation on CD4+ and CD8+ T lymphocyte counts and the CD4/CD8 ratio in treatment-experienced patients are scarce.

Antiretroviral therapy suppresses HIV replication, allowing progressive CD4 T-cell recovery, the continuous normalization of CD8+ lymphocyte T-cells, and a higher CD4/CD8 ratio (>0.9) [18].

Real-life data from cohorts are also available, and efficacy results in maintaining viral suppression were consistent with data from randomized clinical trials at week 48 and week 96. 

This study aimed to compare the real-life data of two multicenter Spanish cohorts of PLWHIV treated with DTG plus 3TC (SPADE-3) or RPV (DORIPEX) as a switch strategy, not only in terms of virological suppression, safety, and durability but also in terms of immune restoration. 

## 2. Material and Methods

### 2.1. Patients and Study Design

We conducted two retrospective, observational, multicenter studies of 638 and 943 virologically suppressed HIV-1-infected patients in two cohorts who switched to 2DRs with DTG plus RPV (from June 2018 to May 2019) or DTG plus 3TC (from August 2018 to August 2021). Thirteen Spanish hospitals integrated the DTG+3TV cohort, extending to 11 other hospitals in the DTG+RPV cohort. All patients fulfilled the following inclusion criteria: (a) treatment-experienced PLWHIV aged ≥ 18 years; (b) switching from 3DRs to DTG-based 2DRs, either with RPV or 3TC at least 48 weeks before the start of the study; and (c) HIV RNA viral load < 50 copies/mL in the previous 24 weeks before switching. Data were collected from medical records, anonymized, and entered into an online electronic database, REDCap [19].

Before starting the study, ethical approval was obtained from central and local ethics committees. Due to the study’s retrospective nature, specific, informed consent was not required. The patients received information about adherence issues and drug reactions in this study when needed.

Data collected included demographics (age, sex, and race); HIV-related data (mode of HIV-1 acquisition); the existence of a prior AIDS-defining illness; HIV treatment status at the time of switching to a 2DR; total time on cART b and the number and type of cART regimens before switching; antiretroviral resistance profile; CD4+ and CD8+ cell counts; HIV-1 viral load (VL); reasons for switching, tolerability, and safety profiles; and non-HIV-related laboratory data, such hepatitis co-infections, pre-existing comorbidities, and laboratory results. In addition, virological failure was confirmed when available by sequencing the pol protein and comparing the relevant mutations to the Stanford and IAS mutation list.

### 2.2. Outcomes

The primary endpoint was the percentage of patients with virological suppression while on DTG plus 3TC and DTG plus RPV (defined as a plasma HIV-1 VL < 50 copies/mL) at weeks 24 and 48.

Secondary outcomes included the following: (a) proportion of patients that experienced the protocol-defined loss of virological control by week 48 (defined as two consecutive HIV-1 VL measurements of >200 copies/mL); (b) changes in the immune status in terms of increase in CD4+ and decrease in CD8+ T lymphocyte counts (cell/mm^3^) as well as improvements in CD4 +/CD8+ ratio (to describe which of these three parameters is more sensitive to changes over time in pre-treated patients); (c) rate, incidence, and reasons for discontinuation of treatment over the 48-week study period; and (d) safety profiles at weeks 24 and 48. 

CD4+ and CD8+ lymphocyte counts were obtained from the patients’ databases at baseline cART with two backbone drugs (abacavir/lamivudine and emtricitabine/tenofovir disoproxil fumarate) and three different third agents (non-nucleoside reverse transcriptase inhibitor = NNRTI, boosted protease inhibitors = bPI, and integrase strand transfer inhibitors = INSTI) at 24 and 48 weeks after switching to dual therapy.

### 2.3. Statistical Analysis

Demographic characteristics, comorbidities, and possible factors associated with HIV infection in the two cohorts of patients were described using descriptive statistics and chi-square tests. 

Association tests were also applied to contrast virological suppression in the two treatment groups. 

Finally, the difference between CD4+, CD8+ lymphocyte count, and CD4/CD8 ratio values between weeks 24 and 48 with the baseline parameter was calculated to estimate possible immunological improvement. The means of these differences were then compared using Student’s *t*-test or the U-Mann Whitney test to see which treatment was more effective.

## 3. Results

### 3.1. Study Population

Overall, 1581 patients were included, of whom 943 (59.6%) were on DTG plus 3TC, and 638 (40.4%) were on DTG plus RPV. Regarding the duration of their treatments, 21.2% of those taking DTG plus 3TC maintained the medication for 24 weeks, 34.9% for 48 weeks, and 44.0% for 96 weeks. Concerning DTG plus RPV, the percentages were 23.2%, 52.8%, and 24.0%, respectively. The baseline characteristics of the study population are described and compared in Table 1, which shows a bivariate analysis in which different demographic characteristics, comorbidities, HIV infection, and possible co-infections are compared individually according to treatment. Ethnicity has been included as a demographic variable (sex and age) to help define the study population. The median age was 50.0 [40.0, 58.0] years in the DTG plus 3TC group and 53.0 [43.0, 58.0] years in the DTG plus RPV group (*p* < 0.001); women represented 23.3% of the participants in the study, and the patients were primarily Caucasian (80.5%). The acquisition of HIV-1 was predominantly through sexual exposure: 58.8 and 69.1% in the DTG plus RPV group and DTG plus 3TC group, respectively. Previous hepatitis co-infections had been diagnosed in 509/1234 of the patients (41.3%), of whom 178 had only the hepatitis B virus (HBV), 182 had only the hepatitis C virus (HCV), and 160 had both hepatitis viruses. Only active co-infections were present in 12 HBV (12/334) patients. In the HBV subgroup, it is necessary to note that they were on the entecavir treatment. 

The median age of HIV diagnosis was 34.0 (25.0, 42.0) years in the DTG plus RPV group and 37.0 (27.0, 47.0) in DTG plus 3TC group. The nadir CD4+T-cell count was 241 cells/μL, and 23.8% of the patients had been diagnosed with AIDS in the DTG plus RPV group and 15.7% in the DTG plus 3TC group. 

Most patients (52.4%) were NNRTI-experienced in the DTG plus RPV group, and (42.2%) had INSTI in the DTG plus 3TC group. 

The most frequent reasons for switching to a DTG-based 2DR were treatment simplification, pill burden, or the number of drugs decreased (67.4%) in the DTG plus RPV and (58.2%) in the DTG plus 3TC groups. Other reasons (toxicity of previous cART regimen, drug–drug interactions, transition therapy to injectable drugs, or cost) were less frequently documented (Figure 1).

At baseline, the median CD4+ lymphocyte count was 701.0 [516.0, 933.0] and 759.0 [556.0, 983.8] cells/μL in the DTG plus RPV and DTG plus 3TC groups. 

### 3.2. Virological Suppression

The rate of virological suppression at weeks 24, 48, and 96 is shown in Table 2. They include the overall population and the various subgroups. At weeks 24, 48, and 96, the virological suppression rates for the overall cohort were 96.9%, 97.4%, and 99.1%, respectively. As shown in Table 2, the suppression is slightly higher in the case of the DTG plus 3TC group. In addition, it can be observed that the percentages of suppression are higher in the cohort that does not present AIDS.

### 3.3. Treatment Discontinuation

The proportion of patients with virological failure over the 48-week study period was 0.01%. Discontinuations per 100 patient-years were 11/940 (1.2%) of the patients in the 3TC group and 8/585 (1.4%) in the RPV group (*p* = 0.981). 

The subsequent genotypic analysis showed no acquired resistance-associated mutations in those experiencing VF.

The most common reasons for discontinuation of the 2DR were the following: treatment changes to another 2DR or a 3DR single-tablet regimen (61.9%), toxicity (18.8%), suitability for future guidelines (8.5%), interaction with other drugs (6.6%), convenience (4.6%), and economic reasons (1.6%). Documented adverse drug reactions (ADRs) were uncommon at the end of the study. Only 1.2% of the patients developed a renal event, 1.0% a neuropsychological event, and 0.4% a digestive event. The DTG plus RPV regimen was found to have greater ADRs in all cases.

### 3.4. Immune Status

Figure 2 and Figure 3 summarize the immunologic status of patients at 24, 48, and 96 weeks of treatment. 

Figure 2 shows the difference in CD4, CD8, and CD4/CD8 values between the different weeks of treatment and the baseline situation in the two treatment regimens. Thus, positive values indicate an increase in the parameter as a function of CD4, CD8, and CD4/CD8. The most significant difference is found in the case of CD4 cells, which increase by 31.5 [−87.8, 128.2], 49.0 [−74.0, 155.0], and 78.0 [−21.5, 189.5], respectively (*p* < 0.001) (Appendix A).

The results in Figure 3 are complementary and align with those described above since more patients with more optimal results are observed for CD4 cells than for the rest of the parameters.

## 4. Discussion

The results of this real-world retrospective, observational, multicenter study support the use of DTG plus 3TC and DTG plus RPV as effective maintenance therapies in virologically suppressed treatment-experienced PLWHIV. DTG plus 3TC and DTG plus RPV provided durable virological suppression and were well tolerated. Despite the differences in baseline characteristics among both groups and a few cases of off-label use, these regimens achieved a rate of virological suppression close to 100% at 96 weeks.

The virological results observed in our study are consistent with those reported in clinical trials. The TANGO study evaluated the efficacy and safety of switching to DTG/3TC from a TAF-based regimen [7] and the randomized pilot clinical trial (ASPIRE), which investigated the efficacy of switching from triple therapy to DTG/3TC. Virological failure (VF) was 0% and virological success was 93% in the former, while VF was 2% and virological success was 91% in ASPIRE. Regarding DTG plus RPV, the results from this study are comparable to those reported in the SWORD-1 and SWORD-2 randomized clinical trials, in which a pooled analysis showed the rate of virological suppression to be 94% [8]. 

Moreover, our results for VF are comparable to the real-world VF reported for DTG-based triple therapy [20,21,22]. However, in this study, the rates of virological suppression varied between 84% and 100% [23,24,25,26,27,28,29,30,31]. Potential factors that may explain the discrepant rates of virological efficacy include differences in the studied populations and the retrospective and observational design of the studies.

The results of this work also support those of previous meta-analyses evaluating both randomized controlled trials and real-world evidence studies, which report a high virological efficacy with DTG-based dual maintenance therapy and a low potential for drug–drug interactions and toxicity [15,32]. 

Overall, in terms of cases of loss of virological control in our study population, there were 43/1357 patients (3.1%) at 24 weeks, 30/1156 patients (2.6%) at 48 weeks, and 5/557 patients (0.9%) at 96 weeks. The resistance analysis showed no acquired resistance-associated mutations. Although sub-optimal adherence cannot be excluded, it is reassuring that the future treatment options for those participants were not compromised. Furthermore, these findings correlate with clinical and real-world trials [6,7,8,23,24,25,26,27,28,29,30,31,33] and show that loss of virological control with DTG-based 2DRs is extremely rare and that the development of resistance to either DTG, 3TC, or RPV is rare. In this sense, our findings are attractive because we can demonstrate high rates of viral suppression despite including long-term diagnosed and pretreated PLWHIV and multimorbidity. Traditionally switching strategies are commonly based on patients undergoing undetectability for at least 24 weeks, and this study gives a new perspective on long-term suppressed and pretreated patients.

The 96-week probability of TF was 1.3% in the RPV group and 0.7% in the 3TC group, which was somehow lower than that expected, based on results of randomized trials [8,34] and observational studies [12,34,35,36,37,38] but consistent with other real-life data [39,40]. In addition, some patients included in this analysis had a long treatment history (the median time since ART start was 22.0 [4.0, 37.0]), had experienced prior VFs, or had detectable resistance mutations. 

Both regimens were well tolerated, with 0.17% of the patients discontinuing treatment. Although we did not observe significant differences in the overall discontinuation rates between the two treatments (0.16 vs. 0.18; *p* = 0.877), the discontinuation rate is slightly lower in the investigational trials, 1−2% in GEMINI 1 and 2, 3% in TANGO [6,7], and 3% in the SWORD-1 and SWORD-2 studies [8]. In several observational studies, the discontinuation rate due to an (adverse reaction) ADR ranged between 2% and 8% for DTG plus 3TC [14,41,42,43] and 2–11% for DTG plus RPV [23,31,44]. Our study’s most frequent adverse event (AE) leading to discontinuation was renal events (1.2%), with only 1.0% neuropsychological toxicity. Real-world data have reported neurotoxicity discontinuation rates of 1–3% for DTG/3TC [41,42,43] and 2–3% for DTG/RPV [43,44]. Given that the observation period of our study was more extensive than all the reports mentioned above, the difference in discontinuation rates observed can most likely be explained by the differences in study populations. Indeed, all of the patients in our study were treatment-experienced. In addition, our study population had been extensively exposed to antiretrovirals before study inclusion and were either less likely to experience AEs or more likely to tolerate AEs, leading to fewer treatment discontinuations.

The incidence of neurotoxicity leading to discontinuation observed in this study is lower than what has been described. This finding argues against an additive or synergistic effect of 2DRs when ADRs appear [45]. 

HBV infection was diagnosed in 27.9% of overall patients, but only 9/177 (5.1%) of diagnosed patients presented a positive surface antigen (HBs Ag) during the switch to DTG+3TC.

Nine patients were newly diagnosed with HBV throughout the DTG/3TC cohort study period, but three were in the DTG/RPV group. These findings highlight the importance of immunizing patients against HBV at 2DR initiation.

A significant increase in the overall mean CD4+ lymphocyte cell count was observed in the DTG/3TC group from baseline to 96 weeks. This increase was independent of sex, comorbidities, or pre-existing AIDS infection stage. In addition, a reduction in the absolute CD8+ value is also observed in the non-AIDS group. In the DTG/RPV group of patients, we observed a significant decrease in the CD8+ lymphocyte T count at week 24 and an increase in the CD4+ lymphocyte T count at week 48. However, we did not observe a change in the mean CD4+/CD8+ ratio in either treatment regimen. This ratio is a marker of immune activation, and a low value is a predictor of non-AIDS-related complications [46]. Similar findings have been reported in patients diagnosed with AIDS. Our data contrast with other published real-life cohort studies, which reported a slight increase in the CD4/CD8 ratio [47], possibly because of the lower baseline CD4/CD8 ratio (0.71) and smaller study population.

The limitations of this study include its retrospective nature, the use of a single-arm analysis which implies the lack of a control group, and publication bias. In addition, due to the retrospective and multicenter design, important data were missing for some of the variables included. Additional limitations of this analysis include those inherent to real-world studies, such as non-randomization, non-registered potential confounding factors in some patients, coding errors, and determination of causality. Likewise, 96 weeks may not be a long enough follow-up to capture some chronic comorbidities.

The strengths of this study include its observational nature, the large sample size with a significant proportion of women, the substantial amount of data collected, the appropriate follow-up time, and, in particular, the diversity of populations, some of whom are typically excluded from RCTs but are, in fact, representative of a real-world setting. In this sense, our results could apply to clinical practice patients.

## 5. Conclusions

Our findings indicate that DTG-based 2DRs (combined with 3TC or RPV) in clinical practice were effective and safe as a switching strategy, with a low VF and high viral suppression rates. Furthermore, the emergence of resistant mutations to DTG, RPV, or 3TC was uncommon, and 2DRs were associated with a favorable immunological recovery. Both regimens were well tolerated, and the ADR rates were low, including neurotoxicity and induced treatment discontinuations.

Thus, a DTG-based 2DRs as a maintenance cART is an excellent option for clinicians to reduce AEs, drug–drug interactions, and costs while preserving antiviral efficacy and providing a high genetic barrier towards resistance development in the large majority of our patients.

## Figures and Tables

**Figure 1 viruses-15-00936-f001:**
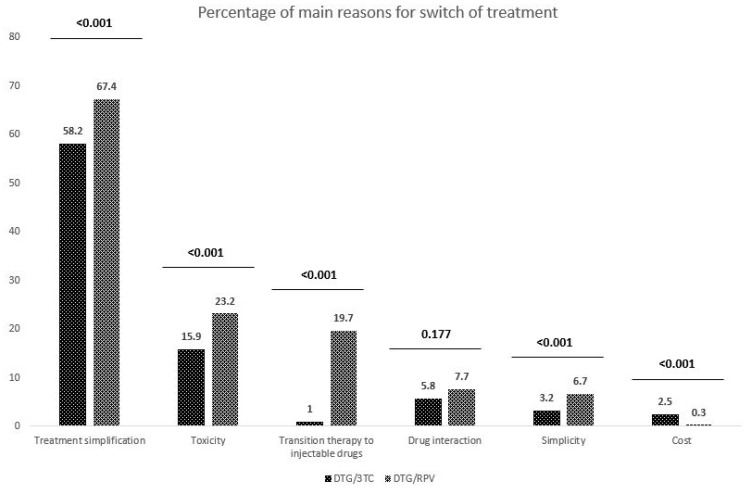
Leading causes of a switch in the DTG plus 3TC and DTG plus RPV cohorts.

**Figure 2 viruses-15-00936-f002:**
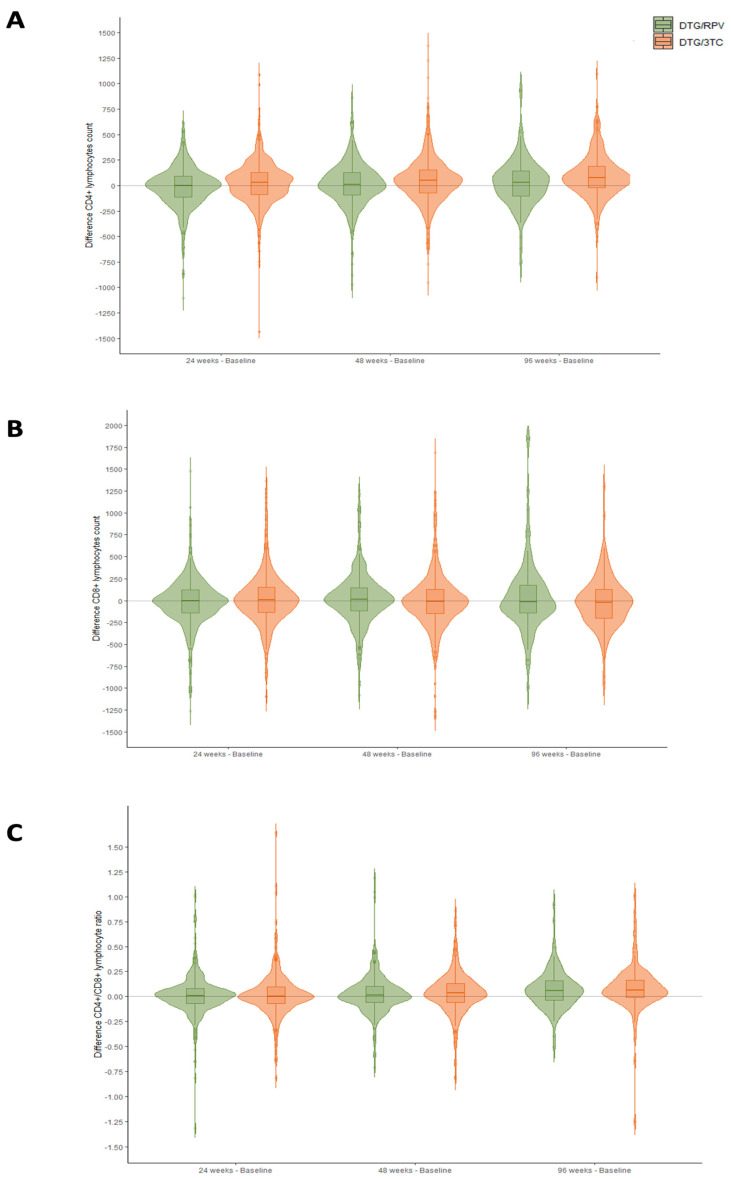
Immunological variation in CD4 T-cell (**A**), CD8 T-cell (**B**), and CD4/CD8 ratio (**C**) parameters between baseline and weeks 24, 48, and 96 of treatment in the DTG plus 3TC and DTG plus RPV cohorts.

**Figure 3 viruses-15-00936-f003:**
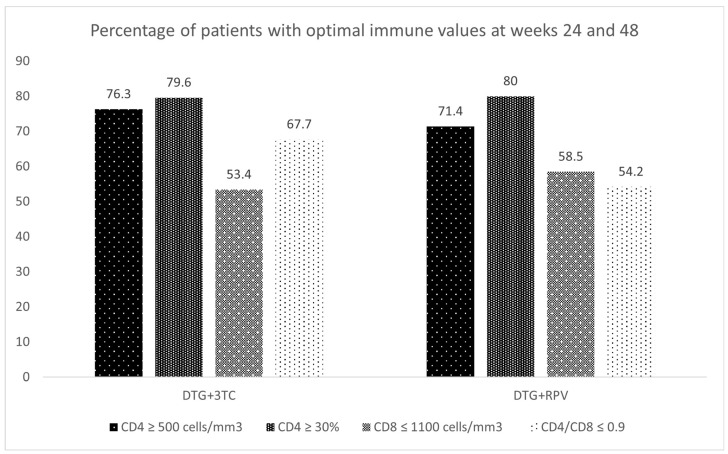
Percentage of patients with optimal immune values at weeks 24 and 48.

**Table 1 viruses-15-00936-t001:** Demographic characteristics, comorbidities, HIV infection, and possible co-infections according to treatment. Comparison between the two regimens.

	DTG Plus 3TC	DTG Plus RPV	*p*-Value
DEMOGRAPHIC			
Age, median [IQR]	50.0 [40.0, 58.0]	53.0 [43.0, 58.0]	<0.001
Male sex n (%)	737/943 (78.2)	475/638 (74.5)	0.099
Spanish nationality n (%)	705/910 (77.5)	534/630 (84.8)	<0.001
COMORBIDITIES n (%)			
Arterial hypertension	109/943 (11.6)	124/638 (19.4)	<0.001
Diabetes	44/943 (4.7)	67/638 (10.5)	<0.001
Dyslipidemia	197/943 (20.9)	198/638 (31.0)	<0.001
Heart Disease	26/943 (2.8)	34/638 (5.3)	0.013
Cerebrovascular disease	9/943 (1.0)	10/638 (1.6)	0.389
Peripheral vascular disease	10/943 (1.1)	13/638 (2.0)	0.168
Kidney failure	37/943 (3.9)	55/638 (8.6)	<0.001
Osteoporosis/Osteopenia	27/943 (2.9)	83/638 (13.0)	<0.001
Chronic pulmonary disease	42/943 (4.5)	55/638 (8.6)	0.001
Psychiatric disorders	74/943 (7.8)	67/638 (10.5)	0.084
Cancer	13/943 (1.4)	17/638 (2.7)	0.099
Chronic liver disease	98/943 (10.4)	94/638 (14.7)	0.012
HIV INFECTION			
Transmission pathways n (%)			
Sexual intercourse	641/923 (69.1)	371/621 (58.8)	<0.001
Intravenous drug injectors	178/923 (19.2)	163/621 (25.8)	<0.001
Immune status, median [IQR]			
Nadir CD4 (cells/mm^3^)	-	283.47 (232.39)	-
Baseline CD4 (cells/mm^3^)	759.0 [556.0, 983.8]	701.0 [516.0, 933.0]	0.003
24 weeks CD4 (cells/mm^3^)	777.5 [596.0, 980.0]	686.0 [509.5, 893.5]	<0.001
48 weeks CD4 (cells/mm^3^)	789.0 [583.5, 1015.5]	702.0 [513.0, 937.0]	<0.001
96 weeks CD4 (cells/mm^3^)	832.0 [608.0, 1059.0]	666.0 [494.0, 939.0]	<0.001
Baseline CD8 (cells/mm^3^)	866.0 [628.5, 1173.5]	839.0 [618.0, 1148.0]	0.392
24 weeks CD8 (cells/mm^3^)	894.0 [647.0, 1203.5]	841.0 [594.0, 1132.0]	0.008
48 weeks CD8 (cells/mm^3^)	889.0 [623.2, 1200.0]	862.8 [644.0, 1119.8]	0.627
96 weeks CD8 (cells/mm^3^)	909.5 [624.5, 1239.2]	890.5 [664.5, 1185.0]	0.868
Baseline CD4/CD8 ratio	0.9 [0.6, 1.2]	0.8 [0.6, 1.2]	0.017
24 weeks CD4/CD8 ratio	0.9 [0.6, 1.2]	0.8 [0.6, 1.2]	0.053
48 weeks CD4/CD8 ratio	0.9 [0.6, 1.3]	0.8 [0.6, 1.2]	0.004
96 weeks CD4/CD8 ratio	0.9 [0.7, 1.3]	0.8 [0.5, 1.1]	0.002
AIDS diagnosis n (%)	759.0 [556.0, 983.8]	701.0 [516.0, 933.0]	0.003
Age of diagnosis, median [IQR]			
Global cohort	37.0 [27.0, 47.0]	34.0 [25.0, 42.0]	<0.001
AIDS patients	46.0 [32.0, 54.0]	36.0 [28.0, 47.0]	<0.001
Non-AIDS patients	34.0 [24.0, 46.0]	33.0 [24.0, 41.0]	0.319
Previous treatment n (%)			
Backbone			
- ABC/3TC	353/943 (37.4)	105/504 (20.8)	<0.001
- FTC/TDF	432/943 (45.8)	126/504 (25.0)	<0.001
- FTC/TAF	137/943 (14.5)	221/504 (43.8)	<0.001
Third agent			
- bPI	246/943 (26.1)	176/638 (27.6)	0.546
- INSTI	435/943 (46.1)	260/638 (40.8)	0.039
- NNRTI	316/943 (33.5)	334/638 (52.4)	<0.001
CO-INFECTIONS, n(%)			
HBV diagnosis	180/631 (28.5)	158/615 (25.7)	0.225
HBsAg positive	9/177 (5.1)	3/157 (1.9)	0.080
HCV positive ELISA	144/635 (22.7)	198/614 (32.2)	<0.001
HCV positive PCR	47/136 (34.6)	55/196 (28.1)	0.135

ABC: abacavir; 3TC: lamivudine; FTC: emtricitabine; TDF: tenofovir disoproxil fumarate; TAF tenofovir alafenamide; BPI: boosted protease inhibitor; INSTI: integrase strand transfer inhibitors; NNRTI: non-nucleoside reverse transcriptase inhibitors; HBV: hepatitis B virus; HCV: hepatitis C virus; HBsAg: hepatitis B surface antigen; ELISA: enzyme-linked immunosorbent assay; PCR: polymerase chain reaction.

**Table 2 viruses-15-00936-t002:** Rate of virological suppression at weeks 24, 48, and 96 by treatment in the overall population, no-AIDS population, and AIDS population.

ALL POPULATION							
	Overall		DTG/3TC		DTG/RPV		
	N	%	N	%	N	%	*p*-Value
24 weeks < 50 copies/mL	1357/1400	96.9	840/860	97.7	517/540	95.7	0.041
48 weeks < 50 copies/mL	1126/1156	97.4	697/711	98.0	429/445	96.4	0.091
96 weeks < 50 copies/mL	552/557	99.1	401/404	99.3	151/153	98.7	0.528
Non-AIDS POPULATION							
24 weeks < 50 copies/mL	887/913	97.2	497/511	97.3	390/402	97.0	0.825
48 weeks < 50 copies/mL	692/710	97.5	366/376	97.3	326/334	97.6	0.823
96 weeks < 50 copies/mL	221/223	99.1	119/121	98.3	102/102	100.0	0.192
AIDS POPULATION							
24 weeks < 50 copies/mL	216/230	93.9	95/98	96.9	121/132	91.7	0.098
48 weeks < 50 copies/mL	177/188	94.1	76/79	96.2	101/109	92.7	0.307
96 weeks < 50 copies/mL	89/89	96.6	38/39	97.4	48/50	96.0	0.710

## Data Availability

All data are kept by the investigators of the SPADE-3 and DORIPEX projects.

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
