# Peer review of "Efficacy and Safety of Two-Drug Regimens with Dolutegravir plus Rilpivirine or Lamivudine in HIV-1 Virologically Suppressed People Living with HIV"

_viruses, 2023, doi:10.3390/v15040936_

Round 1

Reviewer 1 Report

The purpose of this manuscript is to compare the 3DR to 2DR in the aspects of virological suppression, safety, and immune restoration. The findings from the authors are that DTG-based 2DRs are effective and safe as a switching strategy, with a low rate of VF and high rates of viral suppression together with a favorable immunological recovery. This theme is very interesting, and could potentially benefit people living with HIV/AIDS. However lacking controls made the data less convincing. Below are my comments:

1.     People involved in this study are those who have switched from 3DRs to 2DRs. Therefore the seemingly good performance of 2DRs within the period of two years or less may not be due to itself. It can be because of 3DRs previously used that contributed to the good results. My question is can people living with HIV use 2DR directly when they are firstly diagnosed with it? Will the 2DRs have the same efficacy as is shown here in your data?

2.     People who participated in this study have long been exposed to cART before switching to 2DRs. What would be the shortest time when they can switch from 3DRs to 2Drs? (half a year after 3DRs? less than that? Or longer than that?)

3.     Can you define “baseline”? How did you measure it? For example: HIV treatment status at baseline, total time on cART before baseline, number of cART regimens before baseline, previous cART regimen before baseline, baseline CD4, baseline CD8 and baseline CD4/CD8 ratio.

4.     In line 107: The most frequent reasons for starting a DTG-based 2DRs were treatment simplification/pill burden decrease. In line 134: The most common reasons for discontinuation of 2DR were: treatment simplification 134 (61.9%). Can you explain why? The same reason for using and stopping 2DR?

5.     Line 200: Our study's most frequent AE leading to 200 discontinuation were renal events. AE appears the first time here, so please use its full name. Line 162: the same thing for VSS.

6.     Line 239: there is some blank areas, please revise it. Line 241: need to revise it grammatically. Line 29: there is a typo.

Author Response

Review Report 1

  1. People involved in this study are those who have switched from 3DRs to 2DRs. Therefore, the seemingly good performance of 2DRs within the period of two years or less may not be due to itself. It can be because of 3DRs previously used that contributed to the good results. My question is can people living with HIV use 2DR directly when they are firstly diagnosed with it? Will the 2DRs have the same efficacy as is shown here in your data?

In response to the reviewer (line 43), the excellent efficacy and safety of 2DR strategies have led international guidelines to change their recommendations to include dolutegravir + lamivudine [3TC] for treatment-naïve patients. The recommendations are based on the GEMINI 1 and 2 trial results. In this sense, people living with HIV can use dolutegravir + lamivudine directly when diagnosed. The populations studied in this paper defer completely from those ones because they are long-term pre-treated patients with stable HIV status.

  1. People who participated in this study have long been exposed to cART before switching to 2DRs. What would be the shortest time when they can switch from 3DRs to 2Drs? (Half a year after 3DRs? less than that? Or longer than that?)

In response to the reviewer, we have clarified in the paper (line 80) that we included patients who switched to DTG plus RPV or 3TC at least 48 weeks before the start of the study.

  1. Can you define “baseline”? How did you measure it? For example: HIV treatment status at baseline, total time on cART before baseline, number of cART regimens before baseline, previous cART regimen before baseline, baseline CD4, baseline CD8 and baseline CD4/CD8 ratio.

To clarify this aspect, we have changed the word baseline for “at the time of switching to a 2DR”

  1. In line 107: The most frequent reasons for starting a DTG-based 2DRs were treatment simplification/pill burden decrease. In line 134: The most common reasons for discontinuation of 2DR were: treatment simplification 134 (61.9%). Can you explain why? The same reason for using and stopping 2DR?

In response to the reviewer, the most frequent reason for starting DTG-based 2DRs is simplification because most included patients used treatments with more than two pills. There are other classes of 2DR regimens, so you can discontinue this 2DR regimen and use another 2DR regimen. We have included a line with simplification, pill burden, or the number of drugs decreased.

  1. Line 200: Our study's most frequent AE leading to 200 discontinuations were renal events. AE appears the first time here, so please use its full name. Line 162: the same thing for VSS.

As indicated by the reviewer, we have used the full name in AE and VSS and changed some

  1. Line 239: there is some blank areas, please revise it. Line 241: need to revise it grammatically. Line 29: there is a typo.

   Following the reviewer’s comments, we have modified it.

Reviewer 2 Report

Given that DTG is the drug of choice currently to anchor ART regimes and that it is generally used in a three-drug combination, the authors' desire to explore the advantages and disadvantages of reducing this to a two-drug regimen has merit and is of interest to the clinical HIV community.

However, I mostly found the study confusing in its details and I find it difficult based on the way the study is presented to accept the idea that these particular 2-drug regimens provide better virological outcomes for HIV patients. This especially the case as the authors refer to a change as a "switch strategy". (line 36) A patient would change regimens only because of some failure in the previous treatment, whether it be virological, toxicity, adherence, etc. And, these criteria for switching should be elucidated in the inclusion standards. But, I don't see that. 

Ethical Issues and Patient Consent

A second problem that bothers me is the ethical approval language in lines 50 - 52. They state "good clinical practice guidelines were conducted in this study". That is very general language that can cover a wide range of sins. I don't understand exactly what that sentence means. Whose guidelines? what are they? Then they say that consent was not required and ethics approval was obtained before data collection. I find the lack of contact with the patients and obtaining their specific permission troubling. We all would like to avoid needless bureaucratic steps before using data that had previously been arquived, but in this case, those patients have information that could have informed this study about adherence issues and drug reactions.

Inclusion Criteria and Data Collected

In lines 41 - 45, you (changing voice) state that you identified two sets of patients over different time periods and different sets of hospitals? Why? I think it weakens your study and makes it appear that you are hunting patients who will not only meet the switch criteria, but also have the result you desire. Assuming there are central registries of PLWHIV in Spain as your strategy suggests, why not just use 2016 - 2021 and the broadest possible network of hospitals? Also, doesn't relying on hospitals limit the HIV patients that exist in Spain who could have made the switch that interests you.

The data collected (lines 53 - 59), include treatment status at baseline. 2 questions. First, what is the baseline? I don't see a consistent date used for baseline. Second, isn't treatment status a condition of inclusion? If it is, why should there be any variation related to it that needs data to be collected?

Outcomes

 You list various changes in CD4, CD8 and the ratio between them. What are the outcomes you are looking for?

Results

You include a number of IQRs in the text that I could only figure out after I read Table 1. Let the reader know what values they are seeing in the text itself.

Table 1

What does your column for p-value indicate? I take it that you are comparing the two 2-drug regimens, but why? Don't you want to compare them to what the patients did before that justify changing from 3 to 2 drugs?

You have Spanish nationality as a covariate in the Table. Why? What does nationality have to do with the patient's HIV status? Given that Spain is a multi-ethnic society, I didn't understand why I was seeing that. It's a poor analogue these days for ethnicity, if that's what you wanted to measure.

Figure 2

What is the purpose of this given that almost all the violins in all the panels show the same thing? Nice violin plots but the purpose very unclear and what differences in distribution of values you are looking for also unclear.

Resistance Analysis (lines 172 - 174)

What kind of resistance analysis was done? Normally, it would be sequencing the pol protein and comparing the relevant mutations to the Stanford or IAS mutation lists. If you did that, it's an important finding and should be in the paper. If not, how did you do the resistance?

Limitations of Study

I agree that these are limitations. I find the lack of control for confounders troubling because it suggests that you don't know if your results are due to the change in regime, chance or some other factor you haven't measured.

Drug Resistance

On lines 16 - 17, you state "

No emergent resistant virus to dolutegravir has ever 16
been reported in a patient in whom dolutegravir was prescribed in the context of such 17
two-drug regimens

". (Sorry, my paste mechanism did something strange). You cite references 9 and 10 in support. Maybe. I'm not sure that's the point of those 2 articles. You should seek something more on point. And, this claim is definitely not our lab's experience with any type of INSTI-based regimen.

Finally, get a native English speaker to read the text. Your grammar is generally fine, but there are sentences that were tough to figure out.

Author Response

Review Report 2

Given that DTG is the drug of choice currently to anchor ART regimes and that it is generally used in a three-drug combination, the authors' desire to explore the advantages and disadvantages of reducing this to a two-drug regimen has merit and is of interest to the clinical HIV community.

However, I mostly found the study confusing in its details and I find it difficult based on the way the study is presented to accept the idea that these 2-drug regimens provide better virological outcomes for HIV patients. This especially the case as the authors refer to a change as a "switch strategy". (Line 36) A patient would change regimens only because of some failure in the previous treatment, whether it be virological, toxicity, adherence, etc. And these criteria for switching should be elucidated in the inclusion standards. But I don't see that.

Following the reviewer´s comments, we have added and clarified, in the paper, the criteria for switching.

- Ethical Issues and Patient Consent

A second problem that bothers me is the ethical approval language in lines 50 - 52. They state, "good clinical practice guidelines were conducted in this study". That is very general language that can cover a wide range of sins. I don't understand exactly what that sentence means. Whose guidelines? what are they? Then they say that consent was not required, and ethics approval was obtained before data collection. I find the lack of contact with the patients and obtaining their specific permission troubling. We all would like to avoid needless bureaucratic steps before using data that had previously been archived, but in this case, those patients have information that could have informed this study about adherence issues and drug reactions.

As indicated by the reviewer, we have modified the ethical issues and clarified some aspects.

- Inclusion Criteria and Data Collected

In lines 41 - 45, you (changing voice) state that you identified two sets of patients over different time periods and different sets of hospitals? Why? I think it weakens your study and makes it appear that you are hunting patients who will not only meet the switch criteria, but also have the result you desire. Assuming there are central registries of PLWHIV in Spain as your strategy suggests, why not just use 2016 - 2021 and the broadest possible network of hospitals? Also, doesn't relying on hospitals limit the HIV patients that exist in Spain who could have made the switch that interests you.

The data collected (lines 53 - 59), include treatment status at baseline. 2 questions. First, what is the baseline? I don't see a consistent date used for baseline. Second, isn't treatment status a condition of inclusion? If it is, why should there be any variation related to it that needs data to be collected?

- Outcomes

 You list various changes in CD4, CD8 and the ratio between them. What are the outcomes you are looking for?

We decided to compare the changes in CD4, CD8, and CD4/CD8 ratio in both regimens with two objectives: [1] to describe which of these three parameters is more sensitive to changes in long-term pre-treated patients and [2] to visualize the distribution of the difference in each parameter at different weeks of treatment about the baseline situation, considered when patients were switched to these 2DRs. A short line has been added to the main text to clarify this aspect.

- Results

You include a number of IQRs in the text that I could only figure out after I read Table 1. Let the reader know what values they are seeing in the text itself.

The authors consider the IQR well described and referenced in Table 1 and the main text. When we cite a value in the body of the paper, we refer to that value as the median, and it is accompanied by the IQR value with the traditional and usual typography of this parameter. For example: "The median age was 50.0 [40.0, 58.0]". In addition, in Table 1, reference is made to the fact that the value indicated corresponds to the median and the IQR.

Table 1

What does your column for p-value indicate? I take it that you are comparing the two 2-drug regimens, but why? Don't you want to compare them to what the patients did before that justify changing from 3 to 2 drugs?

The p-value in Table 1 corresponds directly to the comparison between the two regimens. This was done because one of the objectives was to contrast the two regimens. We have included in the Table description“ Comparison between the two 2DR regimens.

You have Spanish nationality as a covariate in the Table. Why? What does nationality have to do with the patient's HIV status? Given that Spain is a multi-ethnic society, I didn't understand why I was seeing that. It's a poor analogue these days for ethnicity if that's what you wanted to measure.

In response to the reviewer, we agree that Spain is a multi-ethnic society. However, the ethnicity variable does not act as a covariate. The term covariate is applied when performing some predictive analysis. Table 1 shows a bivariate analysis in which different demographic characteristics, comorbidities, HIV infection, and possible co-infections are compared individually according to treatment. Ethnicity has been included as a demographic variable (sex and age) to help define the study population. We have included this explanation in the main text.

Figure 2

What is the purpose of this given that almost all the violins in all the panels show the same thing? Nice violin plots but the purpose very unclear and what differences in distribution of values you are looking for also unclear.

Thank you very much for the comment regarding the quality of the graphic. The median values are indeed very similar for all violins. However, this is also a result. Also, the median value is one of many pieces of information that can be extracted from this type of graph. It is essential to look at other details, such as the Y-axis's gradation and each violin's width and length. The information that we extract from the graph is the following: [1] There is a more significant margin of variability in the difference of CD8 (note the Y-axis and the length of the violins); [2] CD4 values increase slightly in both patterns, being more noticeable the changes in the DTG plus 3TC pattern (orange violin); [3] the width of the violin represents the density of values. It helps to know the distribution of each variable.

Resistance Analysis (lines 172 - 174)

What kind of resistance analysis was done? Normally, it would be sequencing the pol protein and comparing the relevant mutations to the Stanford or IAS mutation lists. If you did that, it's an important finding and should be in the paper. If not, how did you do the resistance?

We have clarified this aspect in the main text with available data.

Limitations of Study

I agree that these are limitations. I find the lack of control for confounders troubling because it suggests that you don't know if your results are due to the change in regime, chance or some other factor you haven't measured.

We have clarified this aspect in the main text

On lines 16 - 17, you state”.

No emergent resistant virus to dolutegravir has ever been reported in a patient in whom dolutegravir was prescribed in the context of such two-drug regimens". (Sorry, my paste mechanism did something strange). You cite references 9 and 10 in support. Maybe. I'm not sure that's the point of those 2 articles. You should seek something more on point. And, this claim is definitely not our lab's experience with any type of INSTI-based regimen.

Finally, get a native English speaker to read the text. Your grammar is generally fine, but there are sentences that were tough to figure out.

We have checked English writing as we have changed different parts of the paper to a more scientific way.

Reviewer 3 Report

The manuscript by Dueñas-Gutiérrez et al. describes the efficacy and safety of two-drug regimens DTG/3TC in HIV-1 virologically-suppressed PLWHIV. This excellent study will significantly impact controlling drug resistance and maintaining viral suppression in PLWHIV. The article can be accepted after the corrections mentioned below:

1.          Somehow the abstract is missing from the pdf file, and I had to go to the web to read the abstract. Please add it to the main file.

2.          In the abstract, the following sentence needs to be modified to make it more succinct and clear; The most common reason for discontinuation of 2DR were treatment simplification. Adverse drug reactions were uncommon at the end of the study.

3.          All the Figures lack detailed legends.

4.          In Table-2, authors need to define what N stands for. What is the ratio?

5.          In Figure-1 & 3, the author should take better color and shape for the bars to make it clearer for the readers.

6.           Lacks detailed legends in most of the figures.

7.          Authors need to define the abbreviations clearly, e.g., NNRTI.

8.          The article can be accepted after the corrections mentioned above.

Author Response

Review Report 3

  1. Somehow the abstract is missing from the pdf file, and I had to go to the web to read the abstract. Please add it to the main file.

As indicated by the reviewer, we have included the abstract

  1. In the abstract, the following sentence needs to be modified to make it more succinct and clearer; The most common reason for discontinuation of 2DR were treatment simplification. Adverse drug reactions were uncommon at the end of the study.

We have changed the sentence as indicated.

  1. All the Figures lack detailed legends.

We have enlarged the figure legends.

  1. In Table-2, authors need to define what N stands for. What is the ratio?

N means the sample size. Specifically, the proportion is the number of patients who meet the study question (numerator) out of the total number of patients for whom this data is known (denominator). For example, for the first row: 24 weeks < 50 copies/mL, the proportion is 1357/1400. This means that out of 1400 known responses, 1357 patients had <50 copies/mL at 24 weeks.

  1. In Figure-1 & 3, the author should take better colour and shape for the bars to make it clearer for the readers.

The authors consider both the bar type and the colors appropriate. It is a typeface widely used in numerous articles published in high-impact journals. We have not received any suggestions for changes due to reading or interpretation difficulties from the other two reviewers.

  1. Lacks detailed legends in most of the figures.

This is the same comment as number 3. Revised and expanded.

  1. Authors need to define the abbreviations clearly, e.g., NNRTI.

We have defined the abbreviations more clearly

Round 2

Reviewer 1 Report

thanks for answering all my questions. It is now good for publication in this journal

Reviewer 2 Report

You have addressed my previous concerns in the context of a retrospective study.